# A Study on Types of Medication Adherence in Hypertension among Older Patients and Influencing Factors

**DOI:** 10.3390/healthcare10112322

**Published:** 2022-11-20

**Authors:** Sunmin Lee, Kyu-Hyoung Jeong, Seoyoon Lee, Hayoung Park

**Affiliations:** 1Department of Pharmacy, Inha University Hospital, 27 Inhang-ro, Jung-gu, Incheon 22332, Republic of Korea; 2Department of Social Welfare, Semyung University, 65 Semyung-ro, Jecheon 27136, Republic of Korea; 3Interdisciplinary Graduate Program in Social Welfare Policy, Yonsei University, 50 Yonsei-ro, Seodaemun-gu, Seoul 03722, Republic of Korea; 4Institute of Symbiotic Life-TECH, Yonsei University, 50 Yonsei-ro, Seodaemun-gu, Seoul 03722, Republic of Korea

**Keywords:** medication adherence, hypertension, older adult, Latent Profile Analysis (LPA), Korea

## Abstract

Background: Hypertension has the characteristic that the risk of complications can be reduced through appropriate medication in daily life. Hence, it is important to consider practical measures to increase medication adherence, particularly among older patients. Methods: This study used the Korea Health Panel 2020 data (Version 2.0.1), jointly conducted by Korea Institute for Health and Social Affairs and National Health Insurance. A total of 2300 patients with high blood pressure over 65 years of age were selected. In order to identify types of medication adherence in older hypertensive patients, and examine factors that influence the types, the Latent Profile Analysis (LPA) and logistic regression were performed. Results: The types of adherence groups were classified into two groups: an ‘adherence group’ (87.1%) and a ‘non-adherence group’ (12.9%). Furthermore, age, living alone, and depressive symptoms were identified as determinants of medication adherence type among older hypertensive patients. Conclusion: The significant impact of sociodemographic status (age, living alone, and depressive symptoms) on medication adherence among older hypertensive patients indicates the need to establish more specific empirical interventions based on each type’s characteristics. It is expected that this study will provide an in-depth understanding of factors associated with medication adherence among older patients with hypertension, which can support interventions tailored to the specific needs of those who are non-adherent.

## 1. Introduction

Among the most common diseases of old age is hypertension [1]. Although it does not present any symptoms on its own, it is also known as a ‘silent killer’ due to its ability to lead to a variety of complications, such as stroke, renal failure, cerebral infarction, cerebral hemorrhage, and heart disease [2,3,4,5]. It is important to note, however, that despite these risks, hypertension has the characteristic that, unlike other chronic degenerative diseases, the risk of complications can be reduced through pharmacotherapy in daily life [6]. Hence, since pursuing medication adherence has a significant impact on clinical results and the quality of life of high blood pressure patients [7], it is important to consider practical measures to increase medication adherence and maintain proper blood pressure control.

Medication adherence is defined as a degree of consistency of a patient’s behavior with the doctor’s proposal [8], and it is a concept reflecting how well the patient takes the medication prescribed by the doctor at the prescribed frequency, dose, and time [9]. According to a previous study, a patient’s adherence to medication can affect treatment effect and clinical outcome, as well as health-related quality of life [7]. A more comprehensive understanding of patients adherence to medication is essential both in academia and in the field as non-adherence to medication guidelines can result in increased hospitalization or mortality rates or a decrease in the efficiency of healthcare resources [10].

Several factors have been identified as influencing medication adherence through previous studies, including sociodemographic factors such as gender, age, household income, educational level, and region, as well as health-related factors, such as subjective health, frailty, and mental health [11,12,13]. In particular, Kim et al. (2019) found that aging was one of the factors contributing to reduced medication adherence, and the adherence of older patients was lower than that of adults [14,15]. Considering the fact that older patients are highly dependent on and require medical treatment due to the deterioration of physical function, as well as various chronic diseases associated with aging, it is therefore imperative that factors that influence the medication adherence of older patients should be explored.

In spite of the difficulty of measuring medication adherence in older patients, self-reporting medication adherence measurements provide useful information about medication adherence [16,17]. The impact of medication adherence on older adults is multifaceted, but in particular, medication factors are known to have a decisive impact on medication adherence in a variety of ways [18]. In this respect, it is necessary to categorize the multiple forms of taking medication for a strategic approach to increase medication adherence among older patients. There are, however, a limited number of large-scale national studies that can provide a representative sample of older hypertensive patients. Furthermore, since medication adherence is multidimensional, it is possible to assume that there are sub-classes of medication adherence. In order to describe differences between previous studies related to medication adherence, we used latent profile analysis to identify types with common characteristics rather than dichotomous or cluster analysis methods based on the researcher’s arbitrary judgment. 

As a result, this study aimed to categorize the medication adherence of older hypertensive patients using a person-centered approach rather than a variable-centered approach, thereby complementing the limitations of previous studies. Further, this study empirically focused on the demographic and health-related characteristics that were found to be related to medication adherence in previous studies in order to explore the factors affecting medication adherence among older patients with hypertension. This will allow for an in-depth understanding of older patients with hypertension’s medication adherence, as well as suggestions for more specific empirical interventions based on each type’s characteristics. To achieve the purpose of this study, the following research questions have been set.

First, what are the types of medication adherence among older hypertensive patients?

Second, what factors influence the type of medication adherence among older hypertensive patients?

## 2. Materials & Methods

### 2.1. Data

This study aims to identify the type of medication adherence among older hypertensive patients and to examine which factors influence that type of adherence. This study used the Korea Health Panel 2020 data (Version 2.0.1), jointly conducted by Korea Institute for Health and Social Affairs and National Health Insurance. A Korean Medical Panel is a representative panel survey that is designed to identify and evaluate health care service usage behavior and the effectiveness of policies regarding health and medical care [19]. For the analysis of this study, 2300 patients with high blood pressure over 65 years of age who could estimate medication adherence were selected.

### 2.2. Variables 

#### 2.2.1. Independent Variable: Sociodemographic Characteristics

The independent variables were sociodemographic characteristics: gender (male = 0, female = 1), age (continuous variable), equalized annual income (continuous variable), living status (living with someone = 0, living alone = 1), educational background (below elementary school = 1, elementary school = 2, middle school = 3, high school = 4, university or above = 5), residential area (urban = 0, rural = 1), subjective health (very poor = 1, bad = 2, average = 3, good = 4, very good = 5), level of stress (Mostly none = 1, little = 2, a lot = 3, very much = 4), and depressive symptom (no= 0, yes = 1). Particularly, equalized annual income was calculated by multiplying the household annual income by the square root of the number of household members, and then transforming that number into normal distribution by using logarithms. 

#### 2.2.2. Dependent Variable: Medication Adherence (Dose, Dosing Frequency, Time of Administration)

Dependent variables included dose, number of doses, and time of administration. Therefore, medication adherence is measured by asking whether the medication was taken in accordance with the dose, dosing frequency, and timing of its administration. A four-point scale is used to assess medication adherence (not adhered to at all = 1, rarely adhered to = 2, generally adhered to = 3, strictly adhered to = 4). A higher score indicates a higher level of medication adherence.

### 2.3. Statistical Analysis 

M-plus version 8.0 (Muthén & Muthén, Los Angeles, CA, USA)was used to extract a latent group of older hypertensive patients, and SPSS version 27.0 (SPSS Inc., Chicago, IL, USA) was used to verify the factors that affect the type of medication adherence of older hypertensive patients. Following is a detailed description of the data analysis method.

First, in order to confirm the sociodemographic characteristics and medication adherence characteristics of older hypertensive patients, frequency analysis and descriptive statistics were performed using the SPSS 27.0 program. Secondly, Latent Profile Analysis (LPA) was performed on older hypertensive patients using the M-plus 8.0 program. The *p*-value of Akiakie’s Information Criteria (AIC), Bayesian Information Criteria (BIC), Sample-Size Adjusted BIC (SSABIC), Entropy, and Bootstrapped Likelihood Ratio Test (BLRT) were used to determine the optimal number of medical adherence types of older hypertensive patients. The concept of AIC, BIC, and SSABIC are described as information-based fitness indices with a lower value indicating better fit [20]. Entropy is a measure of the average classification accuracy of a model, and it ranges from 0 to 1, with a closer value to 1 representing greater accuracy. BLRT is a method for comparing the k-1 group model and the k group model when there are k latent groups. It indicates that the k-1 group model should be rejected, and the k group model is more appropriate if it is statistically significant. Thirdly, logistic regression analysis was performed with the SPSS 27.0 program to determine the factors affecting the type of medication adherence in older hypertensive patients. This study was conducted with a significance level of 5%. 

## 3. Results

### 3.1. Descriptive Statistics 

The demographic characteristics of the study participants are shown in Table 1. There were 944 men (41.0%) and 1356 female (59.0%), with more female than male. The average age was 73.58 years (Standard Deviation [SD] = 5.68) and the average equivalized annual income was 18,643.12 US dollars (SD = 16,594.67). There were 1729 people living with someone (75.2%), which was about three times more than the 571 people living alone (24.8%). As for the educational background, 277 older adults (12.0%) did not attend elementary school, 936 (40.7%) had completed elementary school, 470 (20.4%) had completed middle school, 459 (20.0%) had completed high school, and 158 (6.9%) had finished higher education above university. In terms of the residential area, 1529 people (66.5%) live in the urban area, which is approximately twice as many as the 771 people (33.5%) who live in the rural area. The average score for subjective health was 2.89 (SD = 0.88) out of 5, and the average score for stress was 1.88 (SD = 0.81) out of 4. 2083 people (90.6%) did not have a depressive symptom, and 217 people (9.4%) did.

In a descriptive statistical analysis of medication adherence, the main variable, the average medication adherence to the dose was 3.89 out of 4 points (SD = 0.34) (Table 2). For the - dosing frequency, the average adherence was 3.86 out of 4 (SD = 0.38), and for the time of administration, it was 3.74 out of 4 (SD = 0.51).

### 3.2. Types of Medication Adherence in Hypertensive Older Patients

AIC, BIC, SSABIC, Entropy, and BLRT were used to evaluate the number of adherence types among older hypertensive patients (Table 3). AIC, BIC, and SSABIC decreased with an increase in the number of types, showing 4 types were confirmed to be the lowest, however, for Entropy, 2 types were higher than those of other types, indicating 0.987. BLRT was significant at the *p* = <0.001 level for all 2, 3, and 4 types of medical adherence. However, 3 and 4 types showed to include a group that represents less than 1% of all cases. Based on the model fit criteria, the 2 types of medication adherence in older hypertensive patients were concluded to be the most appropriate.

As shown in Figure 1, medication adherence was determined via latent profile analysis. For type 1, 2004 patients (87.1%) demonstrated adherence to the dose, dosing frequency, and time of medication, so they were designated as the ‘adherence group’. 296 patients (12.9%) with type 2 showed dose, dosing frequency, and time of medication adherence of around 3 points, so they were categorized as the ‘non-adherence group’.

### 3.3. Analysis of the Study Model 

According to Table 4, factors affecting medication adherence among older hypertensive patients have been identified through logistic regression analysis. A statistically significant fit was found to be appropriate for analysis in this study model (χ^2^ = 45.819, *p* < 0.001). As a result, age (B = −0.054, *p* < 0.001), living status (B = −0.503, *p* < 0.01), and depressive symptoms (B = −0.491, *p* < 0.05) were statistically significant factors affecting medical adherence types among older hypertensive patients. In other words, the younger the age, the more they live with someone, and the less they have depressive symptoms, the greater the probability of belonging to the adherence group. Meanwhile, gender, equalized annual income, educational background, residential area, subjective health, and the level of stress did not show statistical significance to medication adherence.

## 4. Discussion

This study examined which factors influence the type of medication adherence in older hypertensive patients by using the Korea Health Panel. The following are the main analysis results. First, on the basis of the dose, dosing frequency of taking, and time of taking medication, the types of adherence groups were classified into an ‘adherence group’ and a ‘non-adherence group’ [20]. The medication adherence group accounted for 87.1%, which is higher than in previous studies, of older patients taking medication who mostly answered ‘strictly adhered to’ the question asked to observe the dose, dosing frequency, and time of taking medications. The adherence rate to cardiovascular medicines, including hypertensive medicine, has been reported as 46%, 61%, 58%, and 73.9% in previous studies [21,22,23,24]. The ‘non-adherence group’ also reported that the dose, dosing frequency, and time of administration were ‘generally adhered to’, indicating that they relatively adhered to medication well, accounting for 12.9% of older patients. This resulted in consistent adherence with dose amount, number of doses, and time of administration in terms of medication adherence behavior. In contrast, factors related to the number of doses and time taken are known to negatively impact medication adherence [25,26]. It was observed that despite these aspects, the older patients showed relatively fair levels of medication adherence.

Secondly, age, living alone, and depressive symptoms were identified as determinants of medication adherence type among older hypertensive patients. In previous studies comparing the adherence of older patients with their medications based on their age, the results showed inconsistent results due to the diverse research methodologies used [27,28]. This study found, however, that the older the age, the higher the likelihood of belonging to the non-adherent group. The average age of the participants in this study was 73.58 years, which belongs to a relatively older age group among those who are 65 years or older. According to a previous study, analysis indicates that the medication adherence of the older population follows a U-shape, and that non-adherence to medications gradually decreases with age but then increases again after the age of 75 [29]. This could be accounted for by a pattern similar to this study’s findings.

There have been studies showing that the ability to live independently and social care of older adults have a positive effect on medication adherence [30]. In this study, the older population living alone was identified as an influencing factor. Medication adherence may be related to the residential life patterns of these older people. Several follow-up studies on the effect of household-related social care on medication adherence are likely to be necessary, as well as social support plans for managing medication adherence.

The presence of hypertension itself may be considered as a risk factor for the development and incidence of depressive symptoms [31,32]. It has also been demonstrated that depressive symptoms negatively influence medication adherence in hypertensive patients in a number of previous studies, and the results of this study are consistent with these findings [33,34,35,36,37].

In this study, a nationwide survey of about 8500 households and the members of those households was conducted as part of a large-scale research study with national representativeness. Accordingly, this is one of the few large-scale studies to investigate the degree of adherence to medication for hypertension in older patients and to categorize medication adherence behavior. This study is expected to enhance the understanding of medication adherence and its applicability in epidemiological research by evaluating medication use in older patients in Korea.

## 5. Limitation

There are, however, some limitations to this study.

It is important to consider that self-reported medication adherence measurement may overestimate actual adherence [38]. The self-report method is, however, an indirect measurement compared to other medication adherence measure methods such as medication event monitoring systems (MEMS) and pill counts, which have been proven to be highly correlated and are the most effective tools for identifying medication adherence trends [39,40,41]. As we acknowledged, medication adherence research does not only use self-reports; there are other methods as well, however, this study used secondary data that had limited variables, and was only able to use self-reported measurement. In view of the fact that most patients taking blood pressure medications take multiple medications, the pharmacological factors associated with the increase in the number of medications have not been sufficiently examined [42]. Adherence to medications being taken at the same time may be evaluated differently from adherence to hypertensive medications, as well as appearing differently depending on which hypertensive medications are being taken [21,22,43]. Due to the complexity of prescriptions resulting from cardiovascular drugs, follow-up studies are required to adjust and evaluate the overall treatment plan [44]. Moreover, due to the secondary data that had limited variables, we were only able to use limited demographic variables for this study. In this study, the disease severity of the older population was not reflected in the analysis. Cognitive dysfunction is one of the factors that determine medication adherence among older patients, so assessing medication adherence should take cognitive function into account [45,46,47]. It can be concluded that this study has reflected the medication adherence status of older patients who are capable of living a daily life through a survey focusing on healthy community-dwelling general household members without comorbidity or severity.

## Figures and Tables

**Figure 1 healthcare-10-02322-f001:**
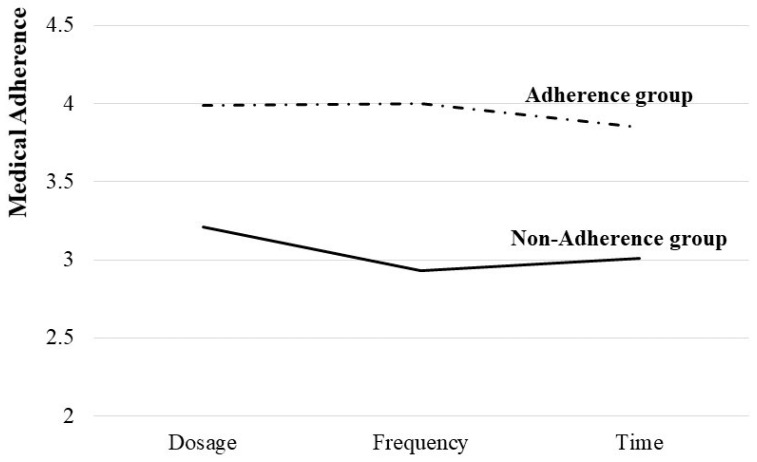
Estimation of Adherence Types.

**Table 1 healthcare-10-02322-t001:** Sociodemographic characteristics of the participants.

(N = 2300)			
Variable	Categories	n	%
Gender	Male	944	41.0
	Female	1356	59.0
Age (M ± S.D)		73.58 (5.68)	
USD $ Equalized Annual Income (M ± S.D)		18,643.12 (16,594.67)	
Living Status	Living with someone	1729	75.2
	Living alone	571	24.8
Educational Background	Below elementary school	277	12.0
	Elementary school	936	40.7
	Middle school	470	20.4
	High school	459	20.0
	University or above	158	6.9
Residential Area	Urban	1529	66.5
	Rural	771	33.5
Subjective health—(M ± S.D)		2.89 (0.88)	
Level of stress (M ± S.D)		1.88 (0.81)	
Depressive Symptoms	No	2083	90.6
	Yes	217	9.4

Note: M = Mean; SD = Standard Deviation.

**Table 2 healthcare-10-02322-t002:** Descriptive statistics of medicine adherence.

(N = 2300)				
Variable	Min	Max	M	SD
Medication adherence—dose	1.00	4.00	3.89	0.34
Medication adherence—dosing frequency	1.00	4.00	3.86	0.38
Medication adherence—time of administration	1.00	4.00	3.74	0.51

Note: Min = Minimum; Max = Maximum; M = Mean; SD = Standard Deviation.

**Table 3 healthcare-10-02322-t003:** Model fit of Latent Profile Analysis.

Class	Model Fit					Groups
	AIC	BIC	SSABIC	Entropy	BLRT *p*-Value	n (%)
1	7119.192	7153.636	7134.573	-	-	-
2	7034.343	7069.142	7051.549	0.987	<0.001	2004 (87.1), 296 (12.9)
3	6984.137	7021.674	7002.967	0.839	<0.001	2004 (87.1), 279 (12.1), 17 (0.7)
4	6910.765	6973.547	6943.531	0.783	<0.001	1986 (86.3), 279 (12.1), 18 (0.8), 17 (0.7)

Note: AIC = Akiakie’s Information Criteria; BIC = Bayesian Information Criteria; SSABIC = Sample-Size Adjusted BIC; BLRT = Bootstrapped Likelihood Ratio Test.

**Table 4 healthcare-10-02322-t004:** Determinants of Medication Adherence Type in Older Patients with Hypertension.

(N = 2300)			
Variables	B	S.E.	Exp (B)
Gender (ref. Male)	0.179	0.148	01.196
Age	−0.054 ***	0.011	0.947
USD $ Equalized Annual Income (log)	−0.200	0.104	0.818
Living Status (ref. living with someone)	−0.503 **	0.147	0.605
Educational Background	−0.058	0.062	0.944
Residential Area (ref. urban)	−0.188	0.135	0.829
Subjective Health	−0.010	0.079	0.990
Level of Stress	−0.080	0.085	0.923
Depressive Symptoms (ref. No)	−0.491 *	0.208	0.612
constant	8.281	1.426	3949.787

* *p* < 0.05, ** *p* < 0.01, *** *p* < 0.001. Note: Coef. = Coefficient; SE = Standard Error; ref = reference.

## Data Availability

The datasets generated during and/or analyzed in this study are publicly available upon request from: https://www.khp.re.kr:444/ (accessed on 1 April 2022).

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
