# Peer review of "A Study on Types of Medication Adherence in Hypertension among Older Patients and Influencing Factors"

_healthcare, 2022, doi:10.3390/healthcare10112322_

Round 1

Reviewer 1 Report

This paper show the types of medication adherence of hypertension among older patients using the whole data in Korea. It is very important to the older patients, so the paper is very necessary. But there are some questions which can be improved in the following. 

1. What is the novel of this paper? Please show it using the appropriate methods or structures. Suggest to show it in the introduction.

2. Suggest to add the impact of  cities on the adherence in Korea. 

3. Please add the suggestion to the institions or practices of other countries or the difference of different countries. 

Author Response

This paper show the types of medication adherence of hypertension among older patients using the whole data in Korea. It is very important to the older patients, so the paper is very necessary. But there are some questions which can be improved in the following. 

  1. What is the novel of this paper? Please show it using the appropriate methods or structures. Suggest to show it in the introduction.

--> We appreciate your suggestion. We have added few sentences to express the novelty of this paper and emphasized it in the introduction.

  1. Suggest to add the impact of cities on the adherence in Korea. 

--> For the cities, we have divided into ‘urban’ and ‘suburban’, however, after thorough discussion among authors, the term ‘rural’ would be more appropriate, therefore, the term ‘suburban’ was replaced with ‘rural’. We apologize for any confusion.

  1. Please add the suggestion to the institions or practices of other countries or the difference of different countries. 

--> We appreciate your suggestion. We have added following paragraph under discussion;

To our knowledge, this is the first study of identifying types of medication adherence in older patients taking antihypertensive medications. The evaluation of medication use in older patients in Korea might increase its applicability and understanding of medication adherence in epidemiological research.

Reviewer 2 Report

The submitted work is well-presented and provides valuable findings, however, there are a few aspects that should be taken into consideration before publishing in the journal of Healthcare. My minor comments and suggestions are given below:

1.     There are plenty of grammatical and sentence structuring mistakes throughout the paper. The authors need another attempt to make it more valuable and well-presented for its readership.

2.     The conclusion part at the end of the abstract needs to be rewritten. The proposed study offers not only a thorough comprehension but also recommendations or proposals based on that knowledge that should be taken into account while taking the present study findings into account. The last statement may be made more insightful by adding a further sentence to it.

3.     The below-given claim at the start of the introduction requires additional literature evidence. One piece of literature evidence is not sufficient enough. Please add at least 3 to 4 recent references to make it more weighted.

Although it does not present any symptoms on its own, it is also known as a 'silent killer' due to its ability to lead to a variety of complications, such as stroke, renal failure, cerebral infarction, cerebral hemorrhage, and heart disease”

4.     Please strengthen the following claim by providing additional literature references.

“When discussing medication compliance, it is necessary to take a comprehensive approach, since dosage, frequency of use, and time of administration have complex effects on adherence.”

5.     Under the statistical analysis section, please mention the significant threshold.

6.     In table 1, there is a typo error for sample size which is written as 2.300 instead of 2300.

7.     The standard expression for categorical data is mean ± S.D. Replace M(SD) with standard representation.

8.     The same should be done in table 2.

9.     The discussion is well-written however limitations should be in a separate heading not merged within the discussion.

Author Response

  1. There are plenty of grammatical and sentence structuring mistakes throughout the paper. The authors need another attempt to make it more valuable and well-presented for its readership.

--> As for your suggestion, we have gone through our manuscript and tried to reduce grammatical errors. We appreciate your comment.

  1. The conclusion part at the end of the abstract needs to be rewritten. The proposed study offers not only a thorough comprehension but also recommendations or proposals based on that knowledge that should be taken into account while taking the present study findings into account. The last statement may be made more insightful by adding a further sentence to it.

--> I appreciate the good feedback you provided. I have modified it to the following statement as you suggested under the abstract.

  1. The below-given claim at the start of the introduction requires additional literature evidence. One piece of literature evidence is not sufficient enough. Please add at least 3 to 4 recent references to make it more weighted.

“Although it does not present any symptoms on its own, it is also known as a 'silent killer' due to its ability to lead to a variety of complications, such as stroke, renal failure, cerebral infarction, cerebral hemorrhage, and heart disease”

--> We appreciate your suggestion. We have provided additional citations in response to your comments.

  1. Please strengthen the following claim by providing additional literature references.

“When discussing medication compliance, it is necessary to take a comprehensive approach, since dosage, frequency of use, and time of administration have complex effects on adherence.”

--> Thank you for pointing out. It has been modified in the main text to reflect the reviewer's point for better understanding.

  1. Under the statistical analysis section, please mention the significant threshold.

--> Thank you for your suggestion. We have mentioned our significance level under 2.3. statistical analysis part.

  1. In table 1, there is a typo error for sample size which is written as 2.300 instead of 2300.

--> We appreciate your detailed comment. We have modified 2.300 into 2,300 in table 1.

  1. The standard expression for categorical data is mean ± S.D. Replace M(SD) with standard representation.

--> We appreciate your detailed comment. We have reflected your suggestion.

  1. The same should be done in table 2.

--> We appreciate your detailed comment. We have modified 2.300 into 2,300 in table 2.

  1. The discussion is well-written however limitations should be in a separate heading not merged within the discussion.

--> Thank you for your suggestion. The limitation section was placed in a separate heading after the discussion part.